# Estimation of Ground Reaction Forces during Sports Movements by Sensor Fusion from Inertial Measurement Units with 3D Forward Dynamics Model

**DOI:** 10.3390/s24092706

**Published:** 2024-04-24

**Authors:** Tatsuki Koshio, Naoto Haraguchi, Takayoshi Takahashi, Yuse Hara, Kazunori Hase

**Affiliations:** Department of Mechanical Systems Engineering, Tokyo Metropolitan University, Tokyo 191-0065, Japan; haraguchi-naoto@tmu.ac.jp (N.H.); t.takahashi0727@tmu.ac.jp (T.T.); hara-yuse@ed.tmu.ac.jp (Y.H.); kazunori.hase@tmu.ac.jp (K.H.)

**Keywords:** biomechanical analysis, human model, contact model, optimization, Kalman filter, ground reaction moment, joint angle, joint torque

## Abstract

Rotational jumps are crucial techniques in sports competitions. Estimating ground reaction forces (GRFs), a constituting component of jumps, through a biomechanical model-based approach allows for analysis, even in environments where force plates or machine learning training data would be impossible. In this study, rotational jump movements involving twists on land were measured using inertial measurement units (IMUs), and GRFs and body loads were estimated using a 3D forward dynamics model. Our forward dynamics and optimization calculation-based estimation method generated and optimized body movements using cost functions defined by motion measurements and internal body loads. To reduce the influence of dynamic acceleration in the optimization calculation, we estimated the 3D orientation using sensor fusion, comprising acceleration and angular velocity data from IMUs and an extended Kalman filter. As a result, by generating cost function-based movements, we could calculate biomechanically valid GRFs while following the measured movements, even if not all joints were covered by IMUs. The estimation approach we developed in this study allows for measurement condition- or training data-independent 3D motion analysis.

## 1. Introduction

In sports focusing on artistic expression (e.g., figure skating, gymnastics, and ballet), rotational jumps involving twists are indispensable for success and high scores. A crucial element in these jumping maneuvers is the force generated during takeoff, determining factors such as jump distance [1] and height [2] as well as midair posture [3]. The analysis of this takeoff force involves the ground reaction force (GRF) and its moment (ground reaction moment, GRM). In particular, in the case of rotational jumps involving twists, conducting a 3D analysis along with joint torques, muscle forces, and ligament forces derived from body movement allows for a control strategy evaluation for optimizing sports movements [4] and injury risks in sports [5,6], among other aspects. Such kinematic indicator measurements often involve the use of a force plate fixed to the ground. While force plates can accurately measure GRFs during movements, their use requires specialized measurement facilities. For successful measurements, the participants have to perform movements with their entire foot on the force plate, potentially leading to unnatural movements. Consequently, such measurements might not accurately reflect the same movements as those performed during actual athletic competitions. In addition, continuous tracking of movements becomes challenging in facilities with no possibility of multiple expensive force plate installations. Moreover, measurement facility establishment could be difficult in places in which equipment installation is possible only underneath the surface (e.g., ice- and snow-covered surfaces or spring-based floors). GRF measurement costs in such environments are substantial, posing further difficulties. To overcome these drawbacks, various lightweight and compact measurement devices have been developed. However, these may not be sufficient for sports motion analysis due to durability concerns, constraints from wearing, and other factors. Wearable force plates save facility installation costs but add weight and height to the shoes, limiting their applicability in high-difficulty movements. Insole-type force sensors address certain challenges but might raise durability and reproducibility concerns, affecting the credibility of their accuracy [7].

To address such challenges, methods have been assessed to measure kinematic elements using only motion capture systems or inertial measurement units (IMUs) and to estimate kinetic elements, such as GRF and GRM [8]. In this study, we focused on using IMUs to estimate GRFs. In measuring sports and daily activities, a small, lightweight, and self-contained measurement method has advantages for clinical applications that avoid movement restriction [9,10]. IMUs meet these conditions, allowing for measurements with minimal location constraints in vast areas (e.g., outdoors; skating rink). Previous studies have explored methods such as using IMUs and machine learning to estimate the three-dimensional (3D) GRFs of gait motion [11] and sagittal-plane GRFs of double-limb jumping [12]. These machine learning-based methods estimate GRFs solely from IMU-recorded motion data, employing a statistical model that incorporates training data collected with both IMUs and force plates. While this approach achieves high accuracy in GRF estimation, it faces challenges in situations in which collecting training data is difficult, such as outdoors or on a skating rink. Another method involves calculating the posture relative to the absolute coordinate system of each body segment from IMU measurements and estimating GRFs through inverse dynamics calculations using the aforementioned values as inputs [13]. However, this method requires a high number of IMUs, thereby potentially hindering movement in sports motion analysis. Therefore, estimation methods that reduce the number of body-mounted sensors attached are crucial for advancing sports motion analysis, in which minimizing sensor interference on natural movement is essential.

In this study, we focused on a GRF estimation method based on forward dynamics calculations using joint torque [14], generating and optimizing body movements using a 3D human model and motion measurement- and internal load-defined cost functions. Using a 3D biomechanical model allows for targeting asymmetrical and nonperiodic sports movements. Moreover, cost function-based movement generation makes biomechanically valid GRF calculation possible while following the measured movements, even if not all joints are covered by IMUs, thereby enabling biomechanically valid GRF estimation with a small number of IMUs. Haraguchi and Hase [14] directly used the acceleration and angular velocity data obtained from the IMUs in this cost function. However, in the case of sports movements, significant noise could be present in the IMU data due to movement intensity. Therefore, in this study, we estimated the 3D orientation using sensor fusion composed of acceleration and angular velocity data obtained from IMUs and an extended Kalman filter, aiming to reduce this noise influence using the estimated values in the optimization calculation.

In this study, we aimed to estimate GRFs and body loads using a 3D forward dynamics approach incorporating sensor fusion with an extended Kalman filter, based on rotational jump movement measurements on land using IMUs.

## 2. Ground Reaction Force Estimation Methods

In this study, we established a biomechanical simulation system that combined forward dynamics simulation using a human model with motion measurement using IMUs. From the forward dynamics simulation-derived generated body movements, the system estimated the GRF, GRM, joint angle, and joint torque. The simulation comprised a forward dynamics model and optimization calculations aimed at cost function minimization, composed of errors between the model and IMU measurements and the internal biomechanical load evaluation of the model. This enabled the model to reproduce biomechanically plausible body movements while following the IMU measurements. Based on the optimized body movements, the GRFs were estimated using a contact model between the human and the ground.

The overall flow of the GRF estimation method was as follows:Height and weight are input into the system and human model construction based on the inertia parameters calculated from them.Based on the reference joint angles formed by the node points, body movement is generated by the forward dynamics model.Body movement evaluation by cost function comprises errors between the generated body movement and measurements from the IMUs and internal biomechanical loads.Repetition of steps 2 and 3 while adjusting the node points of the reference joint angles for cost function value minimization.GRF, GRM, and joint motion estimation using optimized node points in the forward dynamics model.

### 2.1. Forward Dynamics Model

#### 2.1.1. Human Model

Figure 1 illustrates the human model we created, comprising rigid links representing each segment of the body with rotational degrees of freedom (DOFs) defined at the junctions to represent the joints. The segments include the trunk, pelvis, upper arms, forearms, thighs, shanks, and feet, totaling 12 links. The joints exhibit rotational DOFs: three DOFs for right/left hip joint, one DOF for right/left knee joint, one DOF for right/left ankle joint, three DOFs for the lumbar joint, three DOFs for right/left shoulder joint, and one DOF for right/left elbow joints, totaling 21 DOFs. In addition, six virtual DOFs accounted for rotational and translational motion between the pelvis segment and the global coordinate system, bringing the total to 27 DOFs. The knee and elbow joints exhibit flexion-extension, whereas that of the ankle demonstrates plantarflexion-dorsiflexion. The hips, lumbar region, shoulder joints, and pelvic segment are represented by Euler angles in the X (flexion-extension), Y (adduction-abduction), and Z (rotation) sequences. The lengths, inertia moments, and mass centers are determined using estimation equations based on height and weight in Japanese [15,16]. Figure 1 presents how the neutral position for each joint was set and determined by the postural estimation system described in Section 2.2.1. In addition, we defined a sensor coordinate system on the pelvic segment of the human model, allowing for the output of relative acceleration to the global coordinate system, enabling a comparison between the motion generated by forward dynamics simulation and that obtained from the IMU.

In this simulation, motion was generated by calculating joint motion when torque was applied to the joints (defined in Figure 1) using forward dynamics calculations. Figure 2 presents the calculation flow at each time step of the forward dynamics calculation. The forward dynamics model motion equations were as follows:(1)$$M\ddot{q}+\Gamma \left(q,\dot{q},F_{v},F_{h},\tau_{\mathrm{virtual}}\right)=\tau ,$$
where M, Γ, and τ refer to the inertia matrix, vector (consisting of Coriolis forces, centrifugal forces, gravity, and external forces), and joint torque, respectively. q denotes the state variable vector, including the joint angles and the six DOFs between the pelvis and the global coordinate system. Finally, $F_{v}$ and $F_{h}$ are the GRF vectors in the vertical and horizontal directions, respectively, and $\tau_{\mathrm{virtual}}$ represents an external torque vector that prevents the model from falling. We calculated the state variable vector by adding joint torque τ as an input into Equation (1), thereby generating the body movement of the model. We developed the human model using MATLAB/Simulink (10.7) (MathWorks, Inc., Natick, MA, USA) and applied the *ode45* variable-step solver, which implemented the Runge–Kutta method.

#### 2.1.2. Joint Torque Model

The joint torque τ comprises the active and passive torques $\tau_{a}$ and $\tau_{p}$, generated by muscle forces and acting prominently near the joint motion limit range, respectively:(2)$$\tau =\tau_{a}\left(q,\dot{q},q_{r},\dot{q_{r}}\right)+\tau_{p}\left(q,\dot{q}\right).$$
$\tau_{p}$ is represented by a double exponential function that increases the torque near the anatomical joint limits, which are defined by related research [17,18,19,20]. The active torque $\tau_{a}^{j}$ for the joint j, a component of $\tau_{a}$, is calculated using proportional-derivative (PD) control torque based on the joint angle $q^{j}$ for the joint j, a component of q, and reference joint angle $q_{r}^{j}$ for the joint j:(3)$$\tau_{a}^{j}=K_{PD}\left(q_{r}^{j}-q^{j}\right)+D_{\mathrm{PD}}\left({\dot{q}}_{r}^{j}-{\dot{q}}^{j}\right),$$
where $K_{\mathrm{PD}}$ and $D_{\mathrm{PD}}$ are gains in PD control, empirically set to $K_{\mathrm{PD}}=1000\;N\cdot m/rad$ and $D_{\mathrm{PD}}=10\;N\cdot m/\left(rad/s\right)$. The reference joint angle $q_{r}^{j}$ was determined by the optimization calculation described in Section 2.2.

#### 2.1.3. External Force Model

$F_{v}$ and $F_{h}$ in Equation (1) were defined as external forces acting on the foot of the human model and modeled by the contact model. In the human model, there are challenges in dynamic calculations when both feet are in contact and form a closed-loop structure. In gait analysis, statistical models (e.g., transition and distribution functions for GRF and GRM during both-foot support) are often used based on past experiments and data [11,21]. However, in sports movements, existing data remain limited, and statistical models, such as those used in gait analysis, cannot be defined. Therefore, in this study, we applied a contact model to calculate vertical reaction forces by solving the contact problem between the ground and the foot. We determined the horizontal forces by frictional forces, calculated based on the vertical reaction forces. We set the contact points at 22 points for each foot segment, similar to the model of Haraguchi and Hase [14], including distances from the ankle to the heel, metatarsophalangeal joint, and the tip of the second proximal phalanx, as well as heel and metatarsophalangeal joint width, the width between the first and second proximal phalanx, and the heights from the ground to the ankle and the first proximal phalanx. The vertical GRF $F_{v}^{i}$ at contact point i, a component of $F_{v}$, and the horizontal GRF $F_{h}^{i}$ at contact point i, a component of $F_{h}$, were calculated using Equations (4) and (5).
(4)$$F_{v}^{i}=-K_{G}\left(r_{z}^{i}-r_{z_{0}}\right)-D_{G}{\dot{r}}_{z}^{i},$$
(5)$$F_{h}^{i}=-\mu \left({\dot{r}}_{x}^{i},\;{\dot{r}}_{y}^{i},\mu_{s},\;\mu_{d}\right)F_{v}^{i},$$
where $r_{x}^{i},\;r_{y}^{i},$ and $r_{z}^{i}$ represent the position of the contact point, and $r_{z_{0}}$ is set as the height of the ground ($r_{z_{0}}=0$). In addition, $K_{G}$ and $D_{G}$ are the elastic and viscous coefficients of the ground, respectively. The friction coefficient μ is defined as a function of the penetration velocities ${\dot{r}}_{x}^{i}$ and ${\dot{r}}_{y}^{i}$ at each contact point, along with the static and dynamic friction coefficients $\mu_{s}$ and $\mu_{d}$, respectively. We determined the values for $K_{G}$, $D_{G}$, $\mu_{s}$, and $\mu_{d}$ through trial and error, set to $K_{G}=1.0\times 10^{3}\;N/m,\;D_{G}=1.0\times 10^{4}\;N/\left(m/s\right),\;\mu_{s}=0.7,\;\mu_{d}=0.5$.

Furthermore, to prevent the model from tipping over during the simulation, we defined a virtual torque $\tau_{virtual}$ to constrain the model rotation. In forward dynamics analysis, model balance maintenance is essential. When the model attempts to rotate forward, backward, or sideways related to the point of tipping over, virtual torque is applied to a virtual joint. We set the virtual joint between the mass center position of the pelvic segment and the global coordinate system.
(6)$$\tau_{virtual}=-K_{virtual}\;q_{virtual}-D_{virtual}\;{\dot{q}}_{virtual},$$
where $q_{virtual}={\left[\begin{array}{ccc} q_{x} & q_{y} & q_{z} \end{array}\right]}^{T}$ refers to the virtual joint angle represented by the Euler angles of the pelvis relative to the global coordinates. $K_{virtual}$ and $D_{virtual}$ denote gains determined through trial and error, set as $K_{virtual}={\left[\begin{array}{ccc} 400 & 400 & 0 \end{array}\right]}^{T}\;N\cdot m/rad$ and $D_{virtual}={\left[\begin{array}{ccc} 40 & 40 & 0 \end{array}\right]}^{T}\;N\cdot m/\left(rad/s\right)$.

### 2.2. Optimization Calculation

We generated the body movement of the forward dynamics model based on the reference joint angle $q_{r}^{j}$ used in the joint torque model. Optimization calculations for the reference joint angle are typically performed at each time step of the simulation. By performing optimization calculations at high frequencies over the time steps, the generated motion becomes closer to the measured motion. However, this approach also displays certain limitations, such as requiring significant computation time or being susceptible to measurement errors due to large IMU-related noise during certain time steps. In this study, to generate motions closer to the measured motions while reducing the optimization calculation frequency, we applied a method using reference joint angles optimized not at each time step but over the entire measurement period. We used multiple nodes as simulation inputs and determined their values through optimization calculations to create reference joint angle waveforms by cubic spline interpolation. We determined the reference joint angles $q_{r}^{j}$, as follows:(7)$$q_{r}^{j}(t)=spline(N_{1}^{j},\ldots ,N_{m}^{j},t),$$
where spline represents a spline function defined by multiple nodes $N_{1}^{j},\ldots ,N_{m}^{j}$ and the sampling time t. We determined the number of nodes m through trial and error: each upper body joint (i.e., the shoulders, elbows, and lumbar region) displayed two nodes at the motion start and end, and each leg joint involved in the jump (i.e., the hips, knees, and ankles) contained five nodes evenly spaced from the motion start to its end. The other joints are disposed of three nodes set from the motion start to its end. The gait motion model of Haraguchi and Hase [14] sets node points periodically and symmetrically. However, for asymmetric and nonperiodic movement generation, we set the node points as variable numbers for all joints with DOFs.

We established a cost function to assess forward dynamics simulation-generated body movements, and we performed optimization calculations to minimize this value. In this study, we applied genetic algorithm optimization methods and defined the cost functions as follows:(8)$$I_{\mathrm{all}}=\xi_{1}I_{Q}+\xi_{2}I_{a,\;\mathrm{pelvis}}+\xi_{3}I_{\mathrm{muscle}},$$
where $I_{Q}$ and $I_{a,\;\mathrm{pelvis}}$ represent the cost functions for evaluating errors in the 3D orientations of each segment and the acceleration of the pelvis segment, respectively, between the simulation model and the experimental record by the IMUs, as follows:(9)$$I_{Q}=\sum_{s=1}^{12}\int \left\{{\left(\phi_{\mathrm{model}}^{s}-\phi_{\exp}^{s}\right)}^{2}+{\left(\theta_{\mathrm{model}}^{s}-\theta_{\exp}^{s}\right)}^{2}+{\left(\psi_{\mathrm{model}}^{s}-\psi_{\exp}^{s}\right)}^{2}\right\}dt,$$
(10)$$I_{a,\mathrm{pelvis}}=\int \left\{{\left(a_{\mathrm{model},x}^{\mathrm{pelvis}}-a_{\exp,x}^{\mathrm{pelvis}}\right)}^{2}+{\left(a_{\mathrm{model},y}^{\mathrm{pelvis}}-a_{\exp,y}^{\mathrm{pelvis}}\right)}^{2}+{\left(a_{\mathrm{model},z}^{\mathrm{pelvis}}-a_{\exp,z}^{\mathrm{pelvis}}\right)}^{2}\right\}dt,$$
where $\phi_{\mathrm{model}}^{s}$, $\theta_{\mathrm{model}}^{s}$, and $\psi_{\mathrm{model}}^{s}$ denote the simulated orientations of the *s*th segment with respect to the global coordinates, represented by Euler angles in the *X*, *Y*, and *Z* sequences, determined by the $Q_{model}^{s}$. $\phi_{\exp}^{s}$, $\theta_{exp}^{s}$, and $\psi_{exp}^{s}$, measured orientations of the *s*th segment with respect to the global coordinates, represented by Euler angles in the *X*, *Y*, and *Z* sequences. These Euler angles were determined by the measured 3D orientations $Q_{\exp}^{s}$ expressed by quaternions in the global coordinate system recorded by the IMUs, calculated by the extended Kalman filter, as described in Section 2.2.1. $a_{model,x}^{pelvis}$, $a_{model,y}^{pelvis}$, and $a_{model,z}^{pelvis}$ stand for the simulated pelvis segment acceleration. $a_{exp,x}^{pelvis}$, $a_{exp,y}^{pelvis}$, and $a_{exp,z}^{pelvis}$ represent the measured pelvis acceleration by the IMUs. $Q_{\mathrm{model}}^{s}$, $a_{model,x}^{pelvis}$, $a_{model,y}^{pelvis}$, and $a_{model,z}^{pelvis}$ were calculated after the computation of q, $\dot{q}$, and $\ddot{q}$ based on the forward dynamics calculation. $I_{Q}$ and $I_{a,\;\mathrm{pelvis}}$ aim for alignment with the measurements from the IMUs regarding joint motion and for matching the acceleration related to the impact at the moment the foot makes contact with the ground and the translational motion of the body, respectively. $I_{\mathrm{muscle}}$ defines the cost function for the overall muscle load as follows:(11)$$I_{\mathrm{muscle}}=\sum_{j=1}^{21}\int \left({\left|\sigma_{+}^{j}\right|}^{3}+{\left|\sigma_{-}^{j}\right|}^{3}\right)dt,$$
(12)$$\sigma_{+}^{j}=\left\{\begin{array}{c} 0\;\;\;\;\;\;\;\;\;\;\;\;\;\;\;\;:\tau_{a}^{j}<0\\ \frac{\tau_{a}^{j}}{\tau_{+,\max}^{j}}\;\;\;\;\;\;\;:0\leq \tau_{a}^{j} \end{array}\right.,$$
(13)$$\sigma_{-}^{j}=\left\{\begin{array}{c} \frac{\tau_{a}^{j}}{\tau_{-,\max}^{j}}\;\;\;\;\;\;\;\;:\tau_{a}^{j}<0\\ 0\;\;\;\;\;\;\;\;\;\;\;\;\;\;\;\;\;:0\leq \tau_{a}^{j} \end{array},\right.$$
where $\sigma_{+}^{j}$ and $\sigma_{-}^{j}$ represent the active torques normalized by the maximum positive and negative joint torques $\tau_{+,max}^{j}$ and $\tau_{-,max}^{j}$, respectively, for joint j. For instance, the maximum flexion and extension torques at the hip joint are $\tau_{+,max}$ and $\tau_{-,max}$, respectively, which were determined from previous studies across whole joints at the neck [22], lumbar [23,24,25], shoulder [26,27,28], elbow [29], hip [30,31,32], knee [26], and ankle [33]. $I_{\mathrm{muscle}}$ contributes to generating biomechanically valid movements while reducing the overall physical load. In addition, ξ represents the weight factors for each term of the cost function, determined through trial and error, as follows: $\xi_{1}=10^{3},\;\xi_{2}=10^{3}\;(rad^{2}),\;\xi_{3}=10\;{(\left(m/s^{2}\right)}^{2})$.

#### 2.2.1. Orientation Estimation

IMU orientation estimation algorithms have already been developed that use data from 3-axis accelerometers, gyroscopes, and magnetometers to estimate orientation [34]. This approach relies on the IMU’s measurement of gravitational acceleration. However, accurately estimating sensor orientation during high-acceleration sports activities is challenging. Extracting the gravitational acceleration component from the IMU data becomes difficult under such conditions, leading to compromised estimation accuracy. In addition, they rely on magnetometer-derived magnetic field information, making their use challenging in indoor sports settings prone to magnetic disturbances.

In this study, we estimated 3D orientations using sensor fusion of the 3-axis accelerometer and gyroscope data from IMUs [35,36]. Figure 3 presents the algorithm for orientation estimation during motion, which is based on the relationship between the acceleration outputs of the two IMUs attached to each of the two links, making it insensitive to acceleration magnitudes. This approach effectively mitigates the negative impact of large accelerations during sports activities on orientation estimation. Furthermore, it offers the advantage of enabling sensor orientation estimation indoors without requiring magnetometers.

The acceleration output in an IMU $a_{\exp}^{s}$ attached to the segment was represented as the sum of translational, centripetal, tangential, gravitational, and Coriolis accelerations. In this study, Coriolis acceleration could be neglected due to the constant distance between the joint and the IMU. In addition, the gravitational and translational acceleration could be obtained by subtracting the centrifugal and tangential acceleration from the sensor output. Considering the entire system, IMU sensors provide the same gravitational and translational acceleration. Therefore, by expressing the centrifugal and tangential acceleration vectors of the *s*th segment at sampling time t as $a_{\exp,\mathrm{ct}}^{s}$, we expressed this relationship as in the observation equation of the extended Kalman filter (Equation (15)). Furthermore, we set the state values $x^{j}={\left[\begin{array}{cc} Q_{exp}^{s} & Q_{exp}^{s+1} \end{array}\right]}^{T}$ as the 3D orientations calculated by integrating the angular velocity [37] (Equation (14)). In this study, we described 3D orientations using quaternions unaffected by the gimbal lock. The quaternion represents the real part as the first element and the vector of the imaginary parts as the subsequent elements.
(14)$$x^{s}(t+1)=\left(I+\frac{dt}{2}\left[\begin{array}{cc} \Omega^{s} & O\\ O & \Omega^{s+1} \end{array}\right]\right)x^{s}(t)+w^{s},$$
(15)aexps−aexp,ctsaexps+1−aexp,cts+1=Rs+1sxs(t)aexps+1−aexp,cts+1Rss+1xs(t)aexps−aexp,cts+vs,
(16)$$a_{\exp,\mathrm{ct}}^{s}=\omega_{\exp}^{s}\times \left(\omega_{\exp}^{s}\times l^{s}\right)+{\dot{\omega }}_{\exp}^{s}\times l^{s},$$
where Rs+1s denotes the rotation matrix from segment s to s+1. $\omega_{\exp}^{s}$ refers to the angular velocity vector of the *s*th segment at the sampling time t, expressed as $\omega_{\exp}^{s}={\left[\begin{array}{ccc} \omega_{\exp,x}^{s} & \omega_{\exp,y}^{s} & \omega_{\exp,z}^{s} \end{array}\right]}^{T}$, and $l^{s}$ represents the position vector indicating the *s*th segment-attached IMU location from the proximal joint j. We expressed $\Omega^{s}$ in terms of angular velocities as follows:(17)$$\Omega^{s}=\left[\begin{array}{cccc} 0 & -\omega_{\exp,x}^{s} & -\omega_{\exp,y}^{s} & {-\omega }_{\exp,z}^{s}\\ \omega_{\exp,x}^{s} & 0 & \omega_{\exp,z}^{s} & {-\omega }_{\exp,y}^{s}\\ \omega_{\exp,y}^{s} & -\omega_{\exp,z}^{s} & 0 & \omega_{\exp,x}^{s}\\ \omega_{\exp,z}^{s} & \omega_{\exp,y}^{s} & -\omega_{\exp,x}^{s} & 0 \end{array}\right].$$

In this study, we adapted these orientation estimation systems to adjacent IMUs across each joint. In addition, we set the process noise in the extended Kalman filter to $w^{s}={\left[10^{-3}\;10^{-3}\;10^{-3}\;10^{-3}\;10^{-3}\;10^{-3}\;10^{-3}\;10^{-3}\right]}^{T}$ and the observation noise to $v^{s}={\left[10^{-1}\;10^{-1}\;10^{-1}\;10^{-1}\;10^{-1}\;10^{-1}\right]}^{T}(m/s^{2})$. We set the initial value of the posterior error covariance matrix $P_{t}$ to $P_{t_{0}}=diag\{10^{2},\;10^{2},\;10^{2},\;10^{2},\;10^{2},\;10^{2},\;10^{2},\;10^{2}\}$. We determined these values through trial and error, taking into account the magnitude of noise from the accelerometers and gyroscope sensors.

We determined the initial state values of Equation (14) and the neutral position of the model using the IMU data from the calibration posture and this posture estimation system.

## 3. Experiment

### 3.1. Participants

We included seven and three healthy adult males and females, respectively, in the measurements (average ± standard division (SD) value of body height: 1.69±0.10 m, body weight: 63.3±12.7 kg, age: 23.5±2.5 years). This study was approved by the Ethics Committee of Tokyo Metropolitan University. All participants were provided with written and verbal explanations of the measurement details, and their consent was obtained through signed agreement documents prior to initiation.

### 3.2. Conditions

The jumping movement involved a run-up, followed by a jump with a takeoff using the left leg. The run-up motion was initiated in the direction of the jump, and on the second step of the run-up, a takeoff motion was performed followed by a counterclockwise half-turn jump. We adjusted the run-up distance so that the participants could jump naturally. We determined the jumping distance considering safety and the athletic abilities of the participants, with an average ± SD value of 0.52±0.19 m. All participants familiarized themselves with the movements prior to the experiment.

### 3.3. Measurements

We applied IMUs (TSND151, ATR-Promotions Inc., Kyoto, Japan) to measure 3-axis acceleration and angular velocity (sampling frequency: 1000 Hz, accelerator measurement range: ±16 G (resolution: 0.48 mG), gyroscope measurement range: ±2000 dps (resolution: 0.061 dps)). We attached 12 IMUs to the trunk (1), pelvis (2), upper arms (3), forearms (4), thighs (5), shanks (6), and feet (7) of the participants, positioned as follows: (1) at the upper thoracic vertebrae, (2) at the midpoint of the bilateral posterior superior iliac spines, (3) at a quarter of the distance from the elbow joint along the line connecting the shoulder and elbow joints, (4) at the midpoint along the line connecting the elbow joint and the ulnar styloid process, (5) at a quarter of the distance from the lateral condyle of the femur along the line connecting the greater trochanter and the lateral condyle of the femur, (6) at the midpoint along the line connecting the lateral malleolus and the medial malleolus of the tibia, and (7) at the midpoint along the line connecting the calcaneus and the first metatarsal bone base. We attached two cylindrical rods to the IMUs on the forearms to enclose the radius and ulna and secured all IMUs to each segment with elastic belts.

To validate the simulation, we measured the GRFs using a force plate (TF-4060-D, Tech Giken Co., Ltd., Kyoto, Japan) at a sampling frequency of 100 Hz. In addition, we measured the 3D coordinates of markers attached to the entire body at a sampling frequency of 100 Hz using an optical motion capture system (OptiTrack Flex 3, Natural Point Inc., Corvallis, OR, USA). We aligned marker positions and calibration poses using a full-body musculoskeletal model [38] to calculate joint angles and torques. We processed all measurements with a low-pass filter (Butterworth fourth-order type, −3 dB at 18 Hz).

### 3.4. Data Analysis

We used the IMU-derived measurements as input for the forward dynamics model. To validate the model, we calculated the GRF, GRM, joint angle, and joint torque as reference values using force plates and an optical motion capture system. We computed the GRF and GRM from the force plate measurements. The ankle joint center was the point projected onto the ground as the origin. We performed the calculations in a coordinate system aligned with the horizontal direction of the foot. The ankle joint center was determined as the midpoint between markers on the medial and lateral sides of the heel obtained from the motion capture system. In addition, we normalized these values by the body weight (BW) and the BW and body height (BH) product. Furthermore, using the motion capture system and open-source musculoskeletal modeling software OpenSim 4.4 [39,40], along with the full-body musculoskeletal model [38], we performed inverse kinematics and dynamics calculations to compute joint angles and torques, including virtual joints. We normalized joint torques by the BW and BH product, as well as the results from the moment the stepping foot made contact with the ground until it left the ground.

We assessed the concordance between the estimated and true values using Pearson’s correlation coefficient (ρ), defining weak, moderate, strong, and excellent correlations at ρ≤0.35, 0.35<ρ≤0.67, 0.67<ρ≤0.9, and 0.9<ρ, respectively [41]. Finally, we calculated the root mean square error (RMSE) and the relative RMSE (rRMSE). rRMSE normalized the RMSE by the average values of the two data ranges.

## 4. Results

Figure 4 presents the time-series data of the GRFs ((a) the medial direction, (b) the anterior direction, and (c) the vertical direction) and GRM ((d) the sagittal plane, (e) the frontal plane, and (f) the transverse plane). The red line and area indicate the mean ± SD of the estimation, and the gray line and area present the mean ± SD of the measurement. We observed strong correlations in the vertical GRF (ρ=0.749, RMSE=63.5 %BW,rRMSE=27.4%), moderate correlations in the sagittal GRM (ρ=0.520, RMSE=5.67%BW·BH,rRMSE=61.6%), and weak correlations in the medial GRF (ρ=−0.0326,RMSE=32.1%BW,rRMSE=72.9%), anterior GRF (ρ=−0.206, RMSE=44.5 %BW,rRMSE=158%), frontal GRM (ρ=−0.481, RMSE=9.38%BW·BH,rRMSE=84.0%), and transverse GRM (ρ=0.157, RMSE=168%BW·BH,rRMSE=73.7%).

Figure 5 presents the time-series data of the joint angle ((a–c) hip flexion, abduction, external rotation joint angle, (d) knee flexion joint angle, and (e) ankle dorsiflexion joint angle). The red line and area indicate the mean ± SD of the estimation, and the gray line and area present the mean ± SD of the measurement. Table 1 summarizes the cost function estimation accuracy for the joint angles and torques. We detected strong correlations in the case of the hip, knee, and ankle joint flexion angles. We observed weak correlations for the hip and all joint torque abduction and external rotation angles.

## 5. Discussion

### 5.1. Advantages of the Proposed Approach

In this study, we presented a method that successfully achieved accurate estimation for joint angles (i.e., hip, knee, and ankle flexion angles) with relatively large displacement by optimizing biomechanically plausible flexion angles using forward dynamics calculations and the cost function defined by motion measurements and internal biomechanical loads. The accurate estimation of the flexion joint angles led to the correct computation of the vertical movement of the contact points, resulting in high precision in estimating vertical GRF and sagittal GRM.

A previous study used 17 IMUs to estimate the GRF in jump movements using IMUs and biomechanical models [42]. In addition, machine learning-assisted methods require a substantial amount of experimental data for training [12]. However, employing a hybrid cost function including internal biomechanical loads and less noise data estimated from sensor fusion with an extended Kalman filter, our system requires fewer IMUs and eliminates the need for extensive training data. It enabled the cost-effectiveness of GRF, GRM, and joint movement estimation in sports movement. Our proposed estimation approach significantly reduced the hindrance caused by sensor attachments. This minimized sensor interference with a participant’s natural movements during sports motion analysis. While our current system necessitates attaching 12 IMUs, our method inherently allows for GRF estimation, even without sensors on all body segments. This means that the number of sensors may be further reduced in the future, highlighting the potential of our approach to facilitate the clinical applications of GRF estimation in sports movement analysis.

### 5.2. Accuracy

Although the proposed method achieved relatively high accuracy, further improvement is necessary for practical applications. Previous studies have reported estimation accuracy below 20% BW in RMSE for vertical GRFs during jumping and jogging motions [12,42]. A key challenge in IMU-based GRF estimation lies in the inherent susceptibility to noise during measurement, which requires robust attachment and careful calibration to avoid movement inhibition. In this study, we manually attached 12 IMUs to predefined positions, thereby potentially introducing positional inaccuracy-related errors in the estimated GRFs. Furthermore, segment movements during jumping and noise from soft tissues could have adversely affected the GRF estimations. IMUs measure soft tissue acceleration and angular velocity relative to bones [43]. During the measurements, we selected the attachment positions to minimize soft tissue influence, and we secured the sensors with elastic straps. However, in vigorous activities, such as sports movements, IMUs within soft tissues or the belt could experience significant vibrations, potentially leading to considerable measurement errors.

Vigorous sports movements significantly impact GRF estimation accuracy. Compared to the gait analysis results from Haraguchi and Hase [14], our method exhibited decreased accuracy in estimating anterior and medial GRFs. This can be attributed to the inability to accurately calculate foot–ground contact. In this method, there was a significant estimation error in the hip joint angles of abduction and external rotation. In actual movement, the foot was placed forward and outward on the transverse plane, and it landed while rotating in the direction of the rotation, causing the hip joint angles to start from an adducted and internally rotated position. However, in the simulation, the hip joint angles started from an externally rotated position without abduction, causing the foot to land on the transverse plane aligned with the body axis. Small movements, such as abduction and rotation of the hip joint, are susceptible to noise in the IMU measurements, leading to errors in joint angle estimation and inaccurate foot contact. As a result, the accuracy of the GRF ultimately decreased. A potential solution lies in incorporating a damping effect for acceleration into the observation equations of the extended Kalman filter used for 3D orientation estimation. Accurate modeling of the damping effect of acceleration as it propagates through body segments could reduce calculation errors in the sum of translational and gravitational accelerations obtained by the extended Kalman filter’s observation equations for each link. This has the potential to significantly improve the estimation accuracy of both 3D human body orientations and GRFs.

Furthermore, while Haraguchi and Hase incorporated virtual forces for fall prevention in the lateral direction based on the relationship between the overall center of gravity position and the support base, in this study, we introduced virtual torques corresponding to the virtual joint rotation angles to allow for unstable movement generation, such as jumping with a single foot while preventing falls. The addition of virtual torque to the sagittal and frontal plane rotations of the body could result in limited movements. The anterior GRF and frontal GRM accuracy reduction might be attributed to the force application akin to brakes in those directions through virtual torques. Although the system cannot entirely eliminate virtual torques for fall prevention, their impact can be mitigated. One approach involves incorporating an evaluation of virtual torque into the optimization’s cost function. Additionally, employing control methods that maintain balance through joint torque control, such as those using a computational model with nonlinear model predictive control [44], has the potential to further reduce the reliance on virtual torques for balance.

### 5.3. Limitations

We involved healthy, non-athlete participants in this study. When adapting the system to athletes, errors could potentially occur in the estimated GRFs. The current model relies on inertial parameters determined using height- and weight-based estimation formulas and internal values, such as joint torques and muscle forces. These data were obtained from a population that did not represent athletes. However, when targeting sports movements with athlete participants, the use of participant-specific parameters or the average values of athlete populations might be necessary in this sport [45].

The proposed method utilizes a contact model comprised of contact points based on barefoot geometry. To expand clinical applicability, future development efforts will focus on incorporating footwear models, such as sports shoes.

In this study, focusing on rotational jump movements, we set 55 nodes as optimization calculation parameters. We used these node points to calculate the reference joint angle $q_{r}^{j}$, serving as a simulation input. Increasing the node point number might enable more complex body movement generation and enhance ground reaction force estimation accuracy. However, increasing the node point-represented number of variables in the optimization calculation might negatively impact convergence speed and precision. To reduce measurement and computation costs while improving measurement accuracy and optimization calculation precision, kinematic synergy should be considered. Humans smoothly control the redundant multi-DOF musculoskeletal system through a mechanism known as kinematic synergy, thereby reducing controlled DOFs [46]. This control mechanism flexibly manages joint motion through synergistic groups, allowing for diverse body movement generation. Kinematic synergy incorporation into this system and reference joint angle $q_{r}^{j}$ controlling for groups with synergies might allow for body movement generation and GRF estimation, even with fewer IMUs.

Despite its high vertical GRF accuracy, this system displayed lower anterior GRF and frontal GRM precisions, which are potentially unsuitable for predicting the dynamics of all sports. However, for example, on ice, the anterior GRF tends to be small, and the frontal GRM is diminished due to the narrow blade width [47]. As virtual torque might have possibly contributed to such low GRF estimation accuracies, our method could be applicable to sports involving sliding and like figure skating rotation jump movements, where these values are small. If this estimation method could be applied to figure skating, it would allow for the estimation of forces exerted on the ice surface and physical loads with simple measurements, facilitating their application in movement analysis.

This system completes a single trial in approximately 24 h (CPU: Intel^®^ Core i9-13900KF, with 24 cores, 32 logical processors, and an average speed of 3.0 GHz; memory: 128 GB; software: MATLAB R2023a (9.14); operating system: Windows 11 Pro). Therefore, it is not currently adapted to the real-time analysis of sports movements. To reduce computational costs in future optimizations, minimizing the number of reference joint angles used as search parameters could be effective. While the current method allows independent movement of each joint in the human model, exploiting the natural coordination between joints, like kinematic synergies, has the potential to control human motion with fewer parameters. This approach could significantly reduce the number of required reference joint angles.

## 6. Conclusions

We estimated GRFs and body loads using a 3D forward dynamics approach incorporating sensor fusion with an extended Kalman filter based on rotational jump movement measurements on land using IMUs. By cost function-based movement generation, we could calculate biomechanically valid GRFs while following the measured movements, even if the IMUs did not cover all joints. Therefore, biomechanically valid GRF estimation is possible using a small number of IMUs. To reduce the influence of the significant noise of IMUs due to sports movements in the optimization calculation, we estimated the 3D orientation using sensor fusion composed of acceleration and angular velocity data obtained from IMUs and an extended Kalman filter. This estimation method enabled measurement environment- and machine learning training data-independent 3D motion analysis.

## Figures and Tables

**Figure 1 sensors-24-02706-f001:**
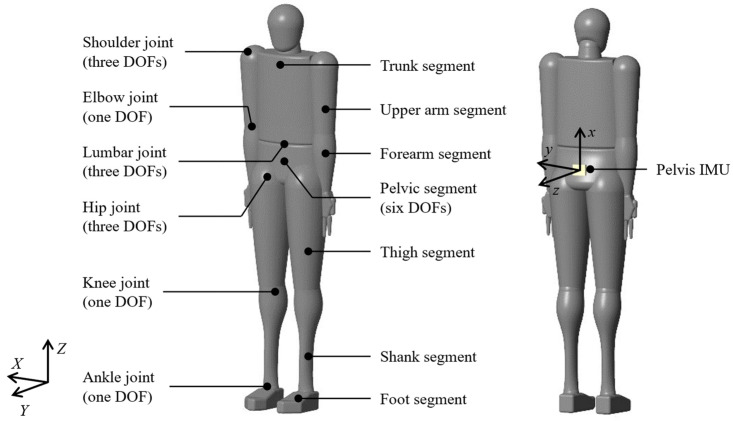
Human body model. The human body model consists of 12 rigid body links representing individual body segments. Joints are modeled using rotational degrees of freedom (DOFs) defined at each link’s connection point. The entire body has 27 DOFs, including a virtual joint with 6 DOFs between the pelvis segment and the global coordinate system. The sensor coordinate system was defined on the pelvic segment of the model and aligned with the orientation of the inertial measurement unit (IMU) used in the experiment.

**Figure 2 sensors-24-02706-f002:**
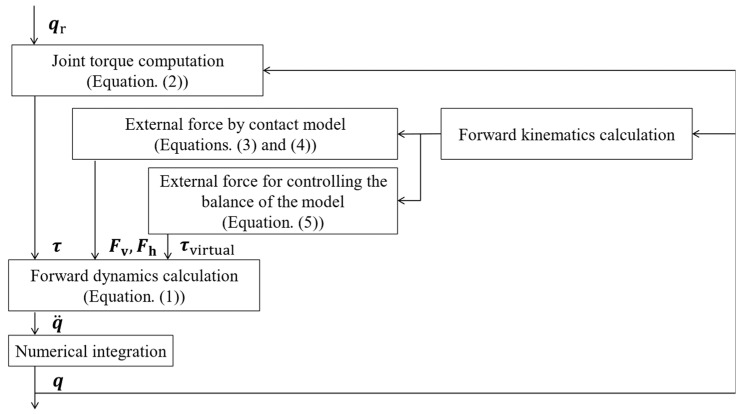
Forward dynamics model flowchart. The model generates body movements through forward dynamics simulation using reference joint angles $\left(q_{r}\right)$ as input. Joint torques (τ), comprised of active ($\tau_{a})$ and passive ($\tau_{p}$) components, are then calculated to determine the joint angles (q). Ground reaction forces ($F_{v}$ and $F_{h}$) and an external balancing force ${(\tau }_{\mathrm{virtual}}$) are obtained from the external force model.

**Figure 3 sensors-24-02706-f003:**
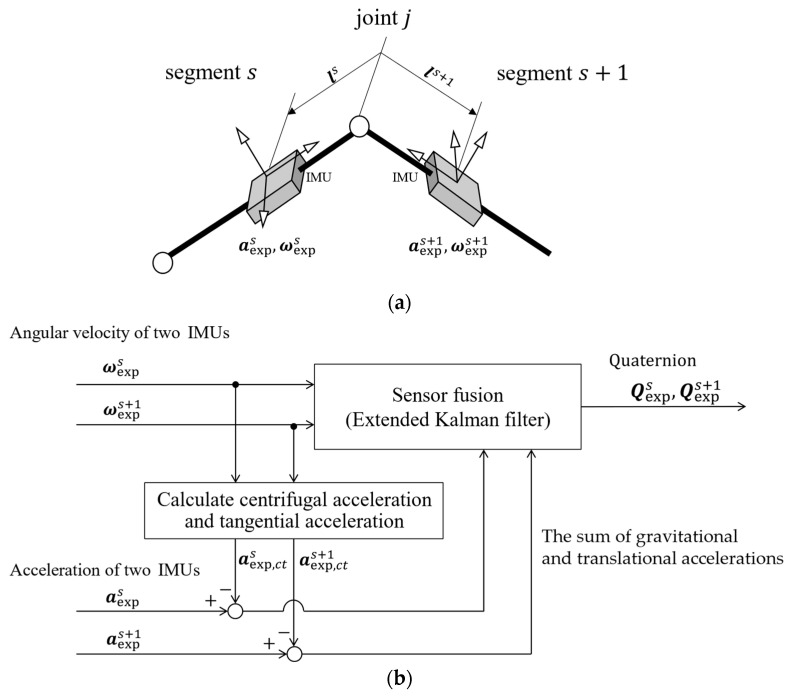
(**a**) Two-link model used for constructing sensor fusion. (**b**) Algorithm for 3D attitude estimation. The IMUs attached to the two links measure accelerations that include translational and gravitational components, which are identical for both sensors. To isolate these components, the centrifugal and tangential acceleration components are subtracted from the IMU outputs, resulting in the sum of gravitational and translational accelerations. This relationship serves as the observation equation in the estimation process. The state values are defined as the angles obtained by integrating the 3D angular velocity. Sensor fusion using an extended Kalman filter is then employed to calculate the 3D sensor orientation (represented as a quaternion) while removing noise, such as drift error.

**Figure 4 sensors-24-02706-f004:**
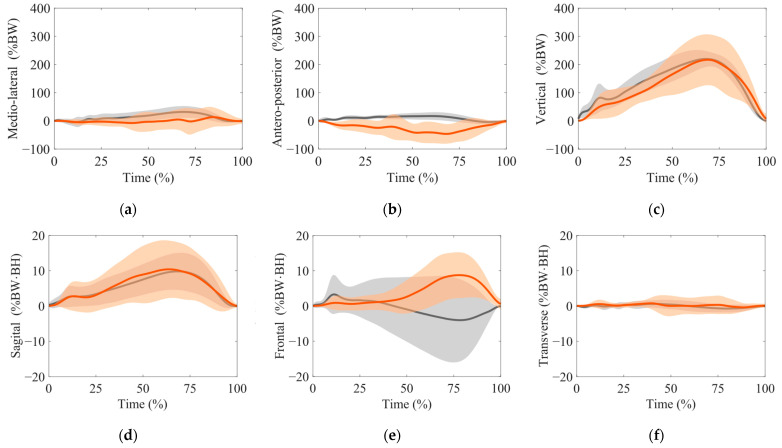
Ground reaction forces (GRFs) (**a**–**c**) and ground reaction moments (GRMs) (**d**–**f**). (**a**) medio-lateral with the medial direction as positive, (**b**) antero-posterior with the anterior direction as positive, (**c**) vertical with the upward direction as positive, (**d**) sagittal plane around the medial axis, (**e**) frontal plane around the anterior axis, and (**f**) transverse plane around the vertically upward axis. These axes originate from the point projected onto the ground from the ankle joint center, and the horizontal direction coincides with the coordinate system of the foot. We defined the timing at the stepping foot heel strike and that at the stepping toe-off as the 0% and 100% phases, respectively. We normalized GRF and GRM by the body weight (BW) and BW × body height (BH), respectively. The red and gray areas and the red and gray solid lines indicate the mean ± standard deviation (SD) of the estimated and measurement values and the mean of the estimated and measurement values, respectively.

**Figure 5 sensors-24-02706-f005:**
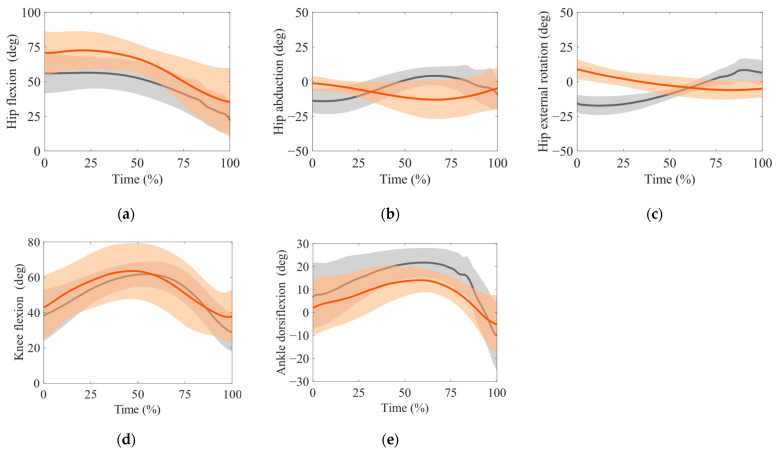
Time-series data of the joint angle. (**a**) hip flexion, (**b**) hip abduction, (**c**) hip external rotation, (**d**) knee flexion, and (**e**) ankle dorsiflexion joint angle. We defined the timing at the stepping foot heel strike and that at the stepping toe-off as the 0% and 100% phases, respectively. The red and gray areas and the red and gray solid lines indicate the mean ± SD of the estimated and measurement values and the mean of the estimated and measurement values, respectively.

**Table 1 sensors-24-02706-t001:** Pearson’s correlation coefficient (ρ), the root mean square error (RMSE), and the relative RMSE (rRMSE) for the joint angles and torques.

			Hip		Knee	Ankle
		Flexion	Abduction	External Rotation	Flexion	Flexion
		ρ				
Joint angle		0.682	−0.291	−0.818	0.781	0.723
Joint torque		0.203	−0.510	0.285	0.376	0.313
		RMSE				
Joint angle	(deg)	20.1	15.9	15.9	15.6	12.3
Joint torque	(BW·BH)	13.7	10.1	3.39	12.2	4.67
		rRMSE				
Joint angle	(%)	56.4	73.7	67.5	43.3	38.6
Joint torque	(%)	55.0	55.1	66.0	42.1	59.5

## Data Availability

The data presented in this study are available on request from the corresponding author.

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
