# Peer review of "Estimation of Ground Reaction Forces during Sports Movements by Sensor Fusion from Inertial Measurement Units with 3D Forward Dynamics Model"

_sensors, 2024, doi:10.3390/s24092706_

Round 1
Reviewer 1 Report
Comments and Suggestions for Authors
Reviewer comments:
This paper presents a complete and clear research framework from the introduction to the conclusion. Through in-depth data analysis and rational discussion, a conceptualization of inertial measurement unit sensor fusion based on a three-dimensional forward dynamics model for estimating ground reaction forces in motion is presented. However, the structure of the article is not compact enough, and the reader cannot easily understand the purpose, methodology and results of the study. It is not well described in some terms and the introduction of the model is not sufficient. There are also many inconveniences in the actual application pathway.
1. Can you elaborate on previous studies that have explored methods for estimating GRFs using IMUs and machine learning?
2. How does the system ensure that the models generated match the actual results?
3. Lines 35-37: For simulation-calculated ligament force metrics that are more indicative of injury risk control strategies, the authors are invited to refer to the relevant literature to supplement their descriptions with appropriate modifications (https://doi.org/10.1016/j.cmpb.2023.107761).
4. Lines 57-59: The reviewer acknowledges the authors' point of view that the use of IMU is uniquely advantageous in solving certain problems. This is relevant literature that can be referenced: https://doi.org/10.5334/paah.313 Application of Nine-Axis Accelerometer-Based Recognition of Daily Activities in Clinical Examination.
3. How does the contact model employed in this study address the challenge of limited existing data for sports movements?
4. How to demonstrate that the method reduces the frequency of optimization calculations?
5. Lines 262-263: How does the proposed method differ from traditional approaches in estimating 3D orientations?
6. The results section is presented too little and may not be sufficiently persuasive, hoping the author will add more of the section.
7. Lines 408-409: What improvements were observed in sagittal GRM estimation as a result of this estimation method?
8. In the Accuracy action, the authors could have added some information about the differences in accuracy between imu and traditional measurements in practical applications. Also, it is important to illustrate the validity of the current methodology by calculating relevant indicators. For example, the authors can refer to previous studies to assess the accuracy of the estimates based on metrics such as RMSE and MSE (https://doi.org/10.1016/j.cmpb.2023.107761).
9. Lines 458-459: It is possible to describe what steps have been taken to reconsider the impact of reducing virtual torque in order to improve the accuracy of body movements and joint torques.
Author Response
We greatly appreciate your comments on our study. Please see the attachment.

Reviewer 2 Report
Comments and Suggestions for Authors
In this paper, the authors proposed a method to estimate GRF and body load using a 3D forward dynamics method that integrates EKF and sensor fusion. Overall, the paper is well structured, but the following points should be clarified to help readers understand.
1. The caption of figure 1 needs to be edited so that it is on the same page as figure 1, and the coordinate system of the sensor of the IMU placed on the pelvis segment must be mentioned in the caption and the coordinate system must be indicated on IMUs sensor in the figure.
2. To help readers understand, it is necessary to specify in lines 376 to 378 of the results that it is time-series data for the measured values (red) and predicted values (gray) of GRF and GRM.
3. Except for vertical and sagittal GRM, the reason why the others have weak correlation is explained in the discussion. However, it is necessary to mention additional methods for minimizing errors.
4. Table 1 needs to be modified to mean that flexion/adduction/external rotation belongs to the angle of the hip.
5. The discussion explains why only the flexion angles of the hip/knee/ankle joint presented as a result in Table 1 show a strong correlation, while other values show a weak correlation. Additional explanation is needed as to why external rotation of hip joints has a strong negative correlation.
6. In estimating GRF in jump movements using IMUs, the authors mentioned that 17 IMUs were used in ref. [40] and said that fewer IMUs were used in this study, but as mentioned in the main text, this study used 12 IMUs, which is believed to be a large number. The authors need to quantitatively describe the effects that can be achieved by reducing the number by five in terms of the amount of computation and amount of learning data.
Author Response

(The authors gave the same response as above.)

Round 2
Reviewer 2 Report
Comments and Suggestions for Authors
All comments raised in round1 was appropriately reflected.